# Diffusion and hydrodynamic instabilities in membrane systems with water solutions of NaCl and ethanol

**Sławomir Grzegorczyn**[1]*, **Iwona Dylong**[1], **Paweł Dolibog**[1], **Andrzej Ślęzak**[2]

**1** Department of Biophysics, Faculty of Medical Sciences in Zabrze, Medical University of Silesia, Katowice, Poland, **2** Collegium Medicum, Jan Dlugosz University, Częstochowa, Poland

* grzegorczyn@sum.edu.pl

**Data Availability Statement:** The data is submitted in the supplementary file: "Raw data for figures. xlsx".

## Abstract

The characteristic manifestations of instability were observed in the form of voltage pulsations measured between electrodes immersed directly in solutions of membrane system chambers, in different configurations of membrane systems. The reason for this type of voltage pulsations is Rayleigh-Benard type instabilities of near-membrane layers caused by density gradients of solutions in these layers. The time of build-up of the concentration boundary layer, after which hydrodynamic instability appears is one of important parameters of these phenomena. The concentration characteristics of these times, measured for one- and two-membrane systems, are nonlinear. With increasing differences in the density of solutions on the membrane at the initial moment, the times of build-up of concentration boundary layers were reduced. In two-membrane systems containing ternary solutions (water, NaCl, ethanol), ethanol was used to control the initial differences in the density of solutions on the membrane. The times of hydrodynamic instabilities in two-membrane system were symmetrical due to the concentration of ethanol, for which the densities of solutions on both sides of the membrane were the same at the initial moment. This dependence is similar for both configurations of the membrane system and is characterized by two nonlinear curves converging to the concentration of ethanol at which, at the initial moment, the densities of the solutions in the chambers of the two-membrane system are the same. In turn, the steady-state voltages of the two-membrane system as a function of the initial concentration of ethanol in the middle chamber with the same initial NaCl concentration in the middle chamber, are a complex function depending on the membrane arrangement. These voltages are characterized by a transition in the ethanol concentration range, for which, at the initial moment, the densities of the solutions in the chambers of the two-membrane system are comparable.

## Introduction

Hydrodynamic instabilities caused by gravity in membrane systems appearing under certain conditions of concentration polarization of a membrane reduce the degree of concentration polarization of the membrane [1]. This increases the transport of the substance and solvent

**Funding:** Sławomir Grzegorczyn BNW-1-048/K/3/I Silesian Medical University, Katowice, Poland funders did not play any role in the study design, data collection and analysis, decision to publish, or preparation of the manuscript.

**Competing interests:** The authors have declared that no competing interests exist.

through the membrane, partially reducing the unfavourable transport conditions of concentration polarization of the membrane [2–4]. The appearance of a sufficiently large solution density gradient at the surface of the membrane directed opposite to the direction of the gravitational acceleration vector is the cause of hydrodynamic instabilities. In addition to the layers near membrane surfaces, gravity-induced hydrodynamic instabilities also occur in Nature in the form of convective motions in the atmosphere [5, 6], as well as in large reservoirs of water where vertical density gradients of saline water occur, caused by temperature differences or gradients of salinity in seawater [7–9].

Another important factor influencing the occurrence of hydrodynamic instabilities in a gravitational field are chemical reactions occurring at the boundary of two reacting and diffusing components, the proportions of which in the solution (substrate concentrations, products concentrations) influence the solution densities [10, 11]. These are dynamic structures of greater complexity than the contact of two liquids with different densities, which results in symmetry breaking of hydrodynamic instabilities appearing under the influence of the occurring chemical reactions [10]. Hydrodynamic instability structures, appearing as a result of solution density disturbances resulting from the diffusion of its components and thus the diffusive change of the solutes concentration fields, are usually characterized by the symmetry of the structure development in the direction of the gravitational field vector, subject to small local fluctuations due to the chaotic nature of the evolution of these structures in time [12]. However, free convection of the solution resulting from hydrodynamic instabilities leads to the equalization of concentrations in solutions (reducing density gradients) and thus to the evolution of the solution towards a homogeneous solution. In both cases, the density gradients associated with the concentrations of individual components are an important driving force of hydrodynamic instability processes [13]. Lack of continuing of reconstruction of these density gradients in the case of pure diffusive mixing of components may lead to more rapid decay of structures resulting from hydrodynamic instabilities than in the case of chemical reactions with diffusion.

In the case of sufficiently large density gradients, directed opposite to the gravitational field vector, buoyancy forces dominate over viscous forces in solutions, causing convective stirring of the medium. This leads to partial elimination of density gradients, also reducing the convective flux of the moving liquid over time. When solution density gradients are reconstructed by external stimuli, such as temperature differences or concentration gradients across the membrane, interaction of these opposing forces leads to a cyclical strengthening and weakening of the convective flow. Setting the conditions of the stimulus reconstructing the density gradients in the medium, e.g. establishing the temperatures of two surfaces—the upper one with smaller temperature and the lower one with larger, the appearance of regular space-time structures known as Rayleigh-Benard convection cells is observed. The nature of these structures also observed in membrane systems with density gradients caused by gradients of solutes concentrations depends on the value of the Rayleigh number [14–16]. For this reason, determining the value of the Rayleigh number in a system where hydrodynamic instabilities are expected is important. Spatial-time structures of hydrodynamic instabilities of various types are also recorded in membrane systems, for example by using interferograms of near-membrane diffusion layers in the Mach-Zehnder method [14, 17]. Various values of critical Rayleigh numbers have to be taken into consideration above which hydrodynamic instabilities caused by gravity appear [18], because they depend on the types of boundaries of the observed volumes with hydrodynamic instabilities, that is rigid or free and fuzzy surfaces. Puthenveettil et al. [19, 20] investigated the transport through a horizontally oriented nylon membrane with a structure of regular square pores. Studying the dynamics of turbulent motions, these authors visualized the structure of plumes formed under conditions of turbulent convection in the range of Rayleigh numbers fulfilling the condition $10^5 < \text{Ra} < 10^{11}$.

The aim of our research was to extend previous research [4, 13, 21, 22] on diffusion and hydrodynamic instabilities in the membrane systems in various configurations of solutions arrangement in relation to the membrane. The membranes were localized in horizontal planes, perpendicularly to the gravitational field vector, so the observed effects had the maximum amplitude. To analyse the evolution of the concentration boundary layers (CBLs) near the membrane, the measurement of the voltage between the Ag|AgCl electrodes immersed directly in electrolyte solutions as a function of time was used [23]. The transport conditions were simulated on the basis of a previously developed model based on the Kedem-Katchalsky membrane transport equations, diffusion equation for CBLs and the Rayleigh concentration number [4, 12, 13]. This model was extended for the multi-component and two-membrane systems. The results obtained from the model were experimentally verified based on measurements for one- and two-membrane systems, with binary and ternary solutions (aqueous NaCl and ethanol solutions). The times of diffusive reconstruction of the CBLs needed to observe gravity-induced hydrodynamic instabilities in CBLs were measured. Based on the experimental data and the developed model, the time and concentration characteristics of observed voltages in the membrane systems and times needed for the appearance of hydrodynamic instabilities in the near-membrane layers were determined. The analysis of these processes occurring in CBLs will allow to expand knowledge about this important phenomenon.

We believe that the knowledge about these phenomena may be important in the consideration of biological systems whose internal environment consists of aqueous solutions of various types of ions, heterogeneous due to the existing boundaries between individual areas, both at the cellular and tissue level. The environment of biological systems is characterized by the presence of many types of ions and non-ionic substances, which concentration gradients can cause solution density gradients. The gravitational field that exist as a permanent factor near the surface of the Earth can influence the cellular and tissue processes of biological systems through hydrodynamic instabilities that appear spontaneously or intentionally. On the other hand, studies of biological systems in zero gravity conditions, in a state of weightlessness [24], show significant changes in the structure and functioning of biological systems. This is probably caused by the disturbance of the previously mentioned processes in the biological system.

Systems with density gradients in weightlessness do not exhibit free convection and thus do not experience hydrodynamic instabilities [25]. Since hydrodynamic instabilities influence transport phenomena in both simple diffusion systems, systems with temperature gradients and complex diffusion systems with chemical reactions or in systems with multifactorial density disturbances (e.g. density gradients caused by different factors at the same time) the differences in the working of technical systems or the functioning of biological systems may be important in a gravitational field in comparison to weightless conditions.

The force of gravity plays an important role both in shaping the universe and cellular biological processes. Research over the past 40 years has shown how exposure to microgravity changes biological processes. It has been shown that microgravity has significant effect on cell proliferation, invasion, apoptosis, migration and gene expression, particularly in cancer cells [26]. These effects can also occur in stem and cancer cells [27]. Therefore, the research can shed light on potential changes in the heterogeneous structures and complex processes of biological systems caused by the gravitational field.

## Theory

One of the phenomenological formalisms describing membrane transport processes is the Kedem-Katchalsky model. This is the model in which homogeneous solutions are assumed on both sides of the membrane and is based on the determination of practical coefficients

characterizing membrane transport. They allow predicting the fluxes of transported substances through membranes. This model can be described by the equations [28]

$$J_v = L_p \left( \Delta P - \sum_{j=1}^{n} \sigma_j \cdot \Delta \pi_j + \beta \cdot I \right) \tag{1}$$

$$J_s = \overline{C}_s (1 - \sigma_s) J_v + \sum_{j=1}^{n} \omega_{sj} \Delta \pi_j + \frac{t_s}{z_s F} \cdot I \tag{2}$$

$$I = -\kappa \cdot \beta \cdot J_v + \kappa \cdot \sum_{j=1}^{n} \frac{t_j}{z_j F} \frac{RT}{C_j} \Delta C_j + \kappa \cdot E \tag{3}$$

where $J_v$ and $J_s$ are the volume and ion fluxes ($s$–indexes for suitable ions, $n$ number of ions in solution), $I$ is the density current through the membrane, $\Delta P$ is the difference of mechanical pressure through the membrane, $\Delta \pi_j$ is the osmotic pressure difference through the membrane for $j$ solute, $\overline{C}_s = (C_h - C_l) \left[ \ln \left( C_h C_l^{-1} \right) \right]^{-1}$ is an average $s$ solute concentration in the membrane and $E = \frac{\Delta \overline{\mu}}{z_s F}$ is the gradient of electrical potential on the membrane. Besides, $C_h$ and $C_l$ ($C_h > C_l$) are the solute concentrations in the chambers at the initial moment, $L_p$, $\sigma_s$ and $\omega_s$ are hydraulic permeability, reflection and solute permeability coefficients for membrane suitably. $\beta$, $t_s$ and $\kappa$ are electroosmotic coefficient, transference number of ions $s$ and conductivity of the membrane suitably. Besides $F$, $R$ and $T$ are the Faraday number, gas constant and absolute temperature suitably, $\overline{\mu}$ is the electrochemical potential of solution and $z_s$ is the valence of ion s.

In turn, diffusion of solute through CBLs can be described by the equation

$$\frac{\partial C_s}{\partial t} = -\frac{\partial J_s}{\partial x} = D_s \cdot \frac{\partial^2 C_s}{\partial x^2} \tag{4}$$

where $D_s$ is the diffusion coefficient of solute $s$ in aqueous solutions.

The characteristic feature of the classical Kedem-Katchalsky model is that it can be only applied to homogeneous solutions, which can be assured by mechanical stirring of the solutions. However, in real conditions (without mechanical stirring of solutions), the phenomenon of concentration polarization of the membrane occurs [3, 4]. This phenomenon is very important from a practical point of view and therefore the Kedem-Katchalsky model needs to be modified [4, 12, 13]. The structures that appear in the membrane system are CBLs, significantly modifying membrane transport [2–4]. For this reason, the description of these structures and their dynamics in connection with the transport of substances through the membrane is of great importance. The combination of the Kedem-Katchalsky equations and the diffusion equation allows for the development of differential equations, giving the possibility of iterative calculation of the time and concentration characteristics of the distribution of substances near the membranes.

Using Eqs (1)–(4) and additional assumptions: $\Delta P = 0$ and $I = 0$, resulting from the conditions of experiment, the transport of substances through membranes in two membrane systems and CBLs near membranes can be described in a new form of difference equations

[4, 13, 22]

$$C_{s,i,n}^{k+1} = C_{s,i,n}^{k} + \frac{\Delta t}{d_w} \cdot \chi_{s,i,n}^{k} - \frac{\Delta t}{d_w^2} D_s \zeta_{s,i,n}^{k} \tag{5}$$

$$\zeta_{s,i,n}^{k} = \begin{vmatrix} B1_s^k \cdot RT \cdot \left( C_{s,1,1}^k - C_{s,2,1}^k \right)(-1)^i & for \ n = 1 \wedge (\ i = 1 \wedge i = 2) \\ 0 & for \ 1 < n < N-1 \\ B2_s^k \cdot RT \cdot \left( C_{s,2,N-1}^k - C_{s,3,1}^k \right)(-1)^{i+1} & for \ (\ n = N-1 \ \wedge \ i = 2) \ \wedge \ (\ n = 1 \wedge i = 3\ ) \end{vmatrix} \tag{6}$$

$$\gamma_{s,i,n}^{k} = \begin{vmatrix} \left( C_{s,i,1}^k - C_{s,i,2}^k \right) & for \ n = 1 \ \wedge \ (i \in \{1,\ 2,\ 3\}) \\ \left( C_{s,2,N-1}^k - C_{s,2,N-2}^k \right) & for \ n = N-1 \\ \left( C_{s,i,n+1}^k + C_{s,i,n-1}^k - 2C_{s,i,n}^k \right) & for \ 1 < n < N-1 \end{vmatrix} \tag{7}$$

where $C_{s,i,n}^k$ are the concentrations of solutes $s$ (first lower subscript) in suitable layers (third lower subscript is for layers $n = 1,2,..$) in the chamber with lower concentration (second subscript $i = 1$ or $i = 3$) and in the chamber with higher concentration (second lower subscript $i = 2$) in a moment of time $k$ (upper subscript). $Bm_s^k = \omega m_s - \sigma m_s \cdot Lm_p(1 - \sigma m_s) \cdot \overline{Cm}_s^k$ ($m$ is an index for membranes: 1 or 2, $s$ for solute) and $\overline{Cm}_s^k = \left( C_{s,2,N-1}^k - C_{s,3,1}^k \right) \cdot \left\{ ln \left[ C_{s,2,N-1}^k \cdot \left( C_{s,3,1}^k \right)^{-1} \right] \right\}^{-1}$ is the average concentration of solute $s$ in membrane $m$. Moreover, $\Delta t$ is a time interval used in the recursive method of solution of Eqs (5)–(7).

Eqs (5)–(7) are used to describe the transport of substances through the membrane with simultaneous reconstruction of CBLs at the membrane surfaces. In order to observe changes in time of CBLs, the method of measuring of voltages between Ag|AgCl electrodes directly immersed into solutions of the membrane system was used. In this case, the voltage between these electrodes can be calculated using the equation

$$\Delta \psi^k = -\frac{2RT}{F} \left[ (\overline{t}_{1+} - t_+) \cdot ln \left( \frac{\gamma_{s,2,1}^k \cdot C_{s,2,1}^k}{\gamma_{s,1,1}^k \cdot C_{s,1,1}^k} \right) - (\overline{t}_{2+} - t_+) \cdot ln \left( \frac{\gamma_{s,2,N-1}^k \cdot C_{s,2,N-1}^k}{\gamma_{s,3,1}^k \cdot C_{s,3,1}^k} \right) + t_+ \cdot ln \left( \frac{\gamma_{s,1,50}^k \cdot C_{s,1,50}^k}{\gamma_{s,3,50}^k \cdot C_{s,3,50}^k} \right) \right] \tag{8}$$

where $\gamma_{s,2,1}^k \cdot C_{s,2,1}^k$, $\gamma_{s,1,1}^k \cdot C_{s,1,1}^k$ and $\gamma_{s,2,N-1}^k \cdot C_{s,2,N-1}^k$, $\gamma_{s,3,1}^k \cdot C_{s,3,1}^k$ are the products of ions activity coefficients (lower index $s$ for Na$^+$ ions) and concentrations suitably at the surfaces of the first and the second membrane and $\gamma_{s,1,50}^k \cdot C_{s,1,50}^k$, $\gamma_{s,3,50}^k \cdot C_{s,3,50}^k$ at electrodes surfaces, located in outside chambers with low NaCl concentrations ($i = 1$ or 3), symmetrically to the membranes, 5 mm from each membrane surface. Besides, $t_+$, $\overline{t}_{1+}$ and $\overline{t}_{2+}$ are the apparent transference numbers for Na$^+$ ions in solution and for membranes 1 and 2 suitably. Solutions in the chambers are homogeneous during mechanical stirring, so at the initial moment of measurement it can be assumed that for all $n$: $C_{s,1,n}^0 = C_l$, $C_{s,2,n}^0 = C_h$, $C_{s,3,n}^0 = C_l$ (initial conditions). As can be seen from Eq (8), the measured voltages are related to the concentrations at the surfaces of the electrodes and membranes. For this reason, this method, by providing an image of time changes in voltage in the membrane system, allows to analyze changes in electrolyte concentrations over time at specific points in the system. Therefore, to ensure that the measured voltage changes reflect changes in CBLs, electrodes were placed 5mm from each membrane in the outer chambers [13]. The diffusive recovery of CBLs and the resulting possibility of

hydrodynamic instability appearance in the vicinity of the membrane affect an important parameter of CBLs, their thicknesses. To determine the CBL thickness, the Lerche criterion was used [29]

$$\frac{C_o - C_\delta}{C_o - C_m} = K \tag{9}$$

where $C_o$, $C_m$ and $C_\delta$ are the solute concentrations in chamber at the initial moment, at membrane surface and at the distance $\delta$ from the membrane suitably. $\delta$ is the thickness of CBL. In our calculations we assumed $K$ equal to 0.01 [17, 29]. After turning off the mechanical stirring of solutions in the membrane system, which initially ensures homogeneity of solutions in chambers of the membrane system, the reconstruction of CBLs follows. The concentrations of individual components permeating through the membrane changes in CBLs in the direction perpendicular to the membrane. If the density of the solution depends on the concentration of its components, the resulting concentration gradient is accompanied by a density gradient of the solution, also directed perpendicularly to the direction of the membrane and parallel to the direction of the gravitational field vector.

If the increasing density gradient is directed like the vector of gravitational field, then the CBLs will be reconstructed only by diffusion [23]. On the other hand, when the density gradient is directed opposite to the gravitational field vector, then in the case of a sufficiently large density gradients, it may force the convective movement of the solution from higher to lower density and in the opposite direction. This is the reason for the appearance of structures at the membrane surface resembling Rayleigh-Benard convective cells or local structures of lower order, such as 'Plum structures' or 'finger structures' [5, 30, 31]. The type of convective motions and their space-time stability are determined by the size of the stimuli on the membrane and the rate of processes supporting the reconstruction of CBLs (diffusion through the membrane) as well as the size of the chambers with solutions and the surface of the membrane. One of the parameters describing the stability of such structures is the Rayleigh concentration number, which can be defined by the equation [32, 33]

$$R_a = \frac{g \cdot \frac{\partial \rho}{\partial C} \cdot \Delta C \cdot \delta^3}{\rho \cdot v \cdot D_s} \tag{10}$$

where $g$ is the gravitational acceleration, $\Delta C$ is the difference of concentrations in CBL, $\delta$ is the thickness of CBL, $\rho$ and $v$ are the density and the kinematic viscosity coefficient of solution suitably, $D_s$ is the diffusion coefficient for solute $s$ in solution.

In the case of membrane systems, the conditions in which hydrodynamic instabilities occur in chambers are determined by the geometry of the membrane system and the CBLs itself. In the case of CBL, where hydrodynamic instabilities may occur, one of the boundaries is the membrane surface, which can be treated as rigid and fixed, while the other boundary of the CBL layer from the solution side is a fuzzy and free. In these boundary conditions of a layer with rigid-fuzzy boundaries, the value of 1100.6 [18] should be assumed as the critical Rayleigh number.

## Materials and methods

One- and two-membrane systems with membranes made of bacterial cellulose (*Biofill* membrane, Fibrocel Productos Biotechnologicos Ltd. Curitiba, Brazil) and polymeric membranes (*Nephrophan* membrane, flat sheet hemodialyzer membrane from cellulose acetate, [triacetate cel-$(O\text{-}CO\text{-}CH_3)_n$], after partial hydrolysis) were used for the study. The membrane systems consisted of two or three cylindrical chambers with volumes 200 cm$^3$ each and surfaces of each

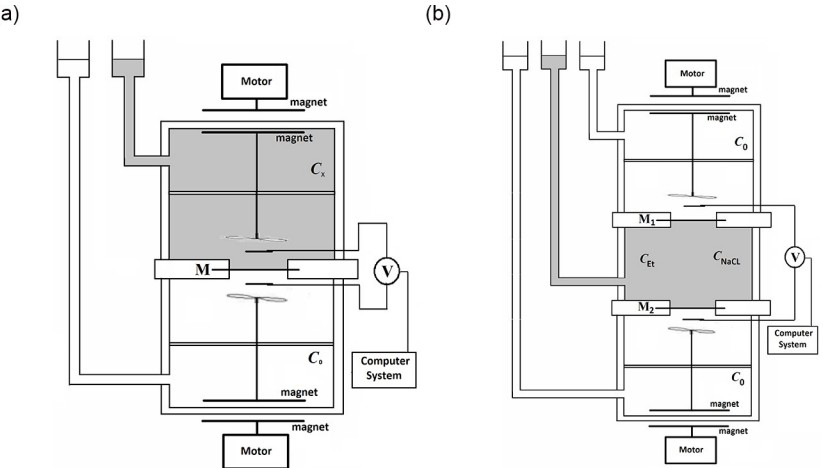

**Fig 1. The one-membrane (a) and two-membrane (b) measurement systems: motor and magnets of stirring system, milivoltmeter (V) connected with Ag|AgCl electrodes and computer system.** NaCl concentrations, homogeneous in whole chambers at initial moment, suitably with higher $C_h$ (grey) and lower $C_l$ (white) NaCl concentrations, membranes were marked as: M, M1 or M2.

membranes separating chambers were $S = 6.1 cm^2$. The membranes were located in horizontal planes. Aqueous solutions of NaCl were used as electrolyte solutions, for which the density of the solution increases with increasing concentration. On the other hand, the addition of ethanol, which is a non-ionic substance, to the solution causes that the increase in ethanol concentration, unlike NaCl, decrease the density of the solution. For this reason, by selecting the appropriate concentrations of ethanol and NaCl at the initial moment, it is possible to control the initial density conditions in the chambers of the membrane system and the conditions in which suitable density gradients appear in the CBLs near the membranes, and thus the possibility of generation of hydrodynamic instabilities caused by the gravity. Fig 1 shows a diagrams of the single-membrane (a) and two-membrane (b) measuring systems. The systems include: a solution stirring systems (motors and magnets), membranes (M, M1, M2) and an electrode system, connected to a computer system. At the initial moment, in the middle chamber of the two-membrane system there was an aqueous solution with higher concentrations of ethanol and NaCl, while in the outer chambers, at the initial moment, there was an aqueous NaCl solution with a concentration 0.01 mol m$^{-3}$. The solutions in the middle chamber were changed depending on the assumed experiment. At the beginning of experiment the NaCl concentration in the middle chamber ranged from 0.5 mol m$^{-3}$ to 50 mol m$^{-3}$. In turn, ethanol was added to the middle chamber to establish the initial differences in the density of solutions on the membranes. In one of the series of experiments in the two-membrane system the initial NaCl concentration in the middle chamber $C_h = 10$ mol m$^{-3}$ was established, while the concentration of ethanol was changed in the range from 0 to 90 mol m$^{-3}$. Before starting the measurement (initial conditions), the solutions were stirred (from 1 to maximal 2 minutes), which ensured the homogeneity of the solutions at the beginning of the measurement. The beginning of the voltage measurement in the systems was the turning off mechanical stirring of solutions, which was also the beginning of diffusive recovery of CBLs near the membrane surfaces.

Cylinder-shaped Ag|AgCl electrodes with a length of 4 mm and a diameter of 0.7 mm were used for the measurements. The electrodes were immersed in the outer chambers of the measuring system, centrally at the membranes, at a distance of 5 mm from the membrane surface, and were connected to a millivoltmeter with a high internal resistance (Meratronik U276,

**Table 1. Basic parameters of *Nephrophan* (polymer membrane) and *Biofill* (Bacterial cellulose membrane) for aqueous NaCl and ethanol solutions, used in the computational model.**

| | *Nephrophan* | | *Biofill* | |
|---|---|---|---|---|
| | **NaCl** | **Ethanol** | **NaCl** | **Ethanol** |
| $L_p \times 10^{11}$ $[\text{m}^3\,\text{N}^{-1}\,\text{s}^{-1}]$ | 0.5 | 0.5 | 6.5 | 6.5 |
| $\omega \times 10^{10}$ $[\text{mol}\,\text{N}^{-1}\,\text{s}^{-1}]$ | 14.3 | 14.5 | 17.1 | 15.3 |
| $\sigma$ | 0.06 | 0.025 | 0.0036 | 0.034 |

internal resistance 0.1 GΩ) connected to a computer. The entire measuring system was placed in a grounded metal chamber (Faraday cage) in order to shield the measuring system from the influence of external electric fields. The membrane systems were thermally stabilized (T = 295K). The solutions preparation error was lower than 1%, while the accuracy of the measured voltage was 0.1 mV. The transport parameters of the membranes (*Nephrophan*–polymer membrane–N, and *Biofill*–bacterial cellulose membrane—CB) used in the experiment are listed in Table 1 and were previously experimentally measured [34]. In turn, diffusion coefficients in water were assumed for NaCl as $D_{\text{NaCl}} = 1.47 \times 10^{-9}$ m$^2$/s and for ethanol $D_{\text{et}} = 1.04 \times 10^{-9}$ m$^2$/s. Experimental data was processed with the aid of MathCad Prime 3.0 and Origin Pro 2022 software.

## Results and discussion

The stability conditions of CBLs in the membrane systems are considered for aqueous solutions of ethanol and NaCl, which densities depend on the concentrations of both components. An increase in NaCl concentration increases the density of the solution, while an increase in ethanol concentration decreases the density of the solution. These relationships can be described by the equation

$$\rho = \rho_o \left(1 + \alpha_{NaCl}\, C_{NaCl} + \alpha_{et}\, C_{et}\right) \tag{11}$$

where the coefficients $\alpha_{\text{NaCl}}$ and $\alpha_{\text{et}}$ were measured experimentally

$$a_{\text{NaCl}} = (4.14 \pm 0.03) \times 10^{-5}\ \text{m}^3\ \text{mol}^{-1},\ a_{\text{et}} = (-0.84 \pm 0.02) \times 10^{-5}\ \text{m}^3\ \text{mol}^{-1}.$$

In membrane systems, the hydrodynamic instabilities lead to voltage pulsations between the electrodes located at a short distance from the membrane surfaces, so that a membrane or two membranes are located between the electrodes. In the case of one-membrane systems, there are configurations of the membrane system with the membrane in a horizontal plane, with a higher density solution above the membrane and with an appropriately high initial concentration above the membrane. Fig 2 presents the initial voltages (during stirring of solutions in chambers) measured in one- and two-membrane systems as a functions of the initial higher NaCl concentrations for the systems: one-membrane systems in configurations with higher NaCl concentrations over the membrane: with *Nephrophan* (●, N), with *Biofill* (●, CB) and two-membrane system with *Nephrophan* and *Biofill* membranes (●, N and CB), where the solid line passing through the points is the difference between the voltages in one-membrane systems with *Nephrophan* and *Biofill* membranes respectively (U$_{\text{CB-N}}$ = U$_{\text{N}}$−U$_{\text{CB}}$).

The measured voltages for homogeneous solutions in membrane systems depend on the concentrations at the electrodes surfaces and the apparent transference numbers of ions for the membrane [21]. The apparent transference numbers of the appropriate ions for the

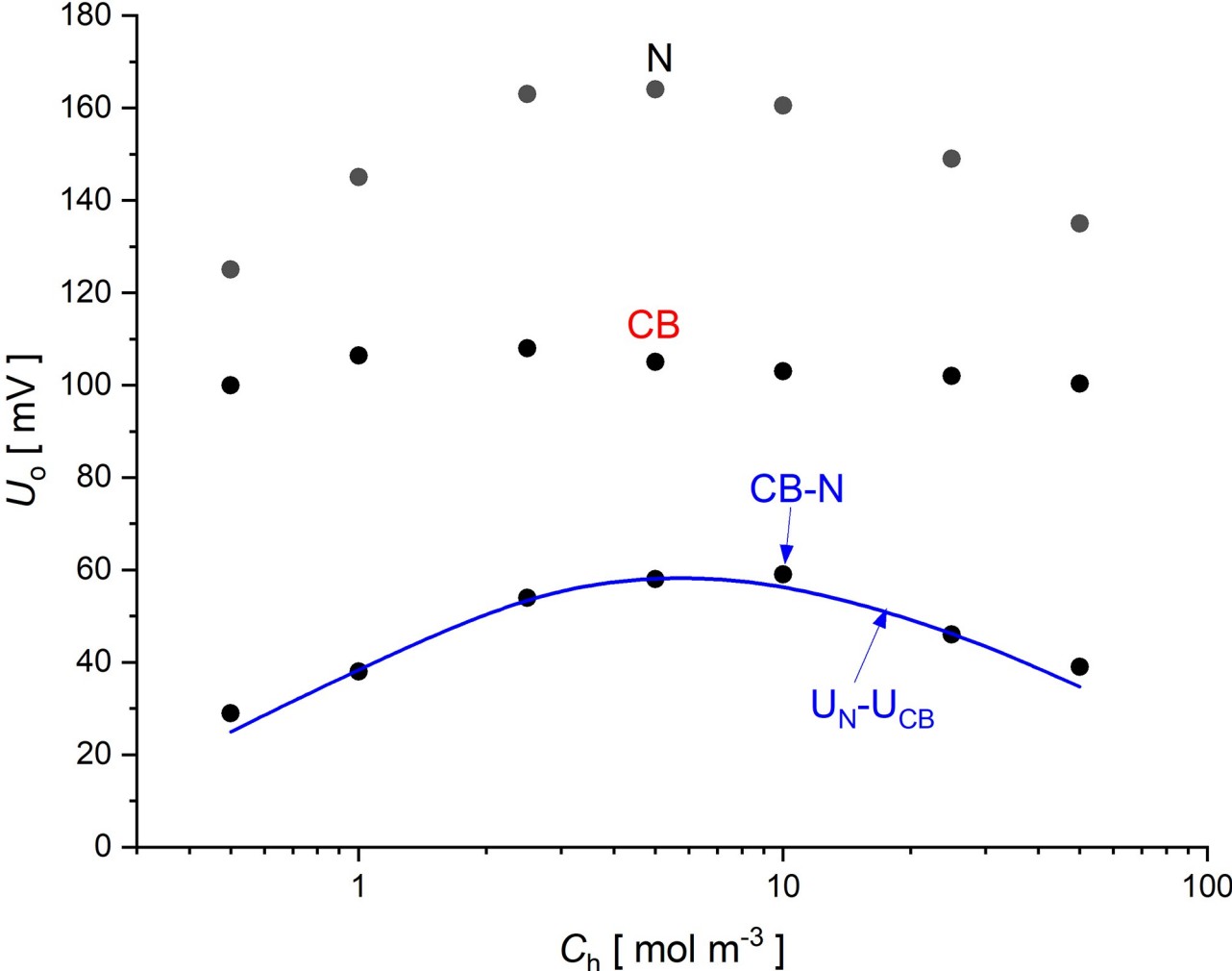

**Fig 2. Voltages for homogeneous solutions in one- and two-membrane systems and aqueous NaCl solutions as a function of the initial higher NaCl concentrations for a one-membrane system with the *Nephrophan* membrane (●, N), for a one-membrane system with a *Biofill* membrane (●, CB) and for a two-membrane system with *Nephrophan* and *Biofill* membranes (●, CB-N).** The solid line is the difference between the voltages in one-membrane systems respectively: ($U_N$−$U_{CB}$).

membrane depend on the ratio of the concentrations on the membrane and the interaction of ions moving in the membrane [13], which causes complex dependencies of the observed voltages on the concentrations of homogeneous solutions in the chambers. Fig 2 shows that at the initial moment, when the influence of CBLs at the membranes can be neglected, the observed voltages in the two-membrane system can be calculated from the voltage difference for one-membrane systems with appropriate membranes and the electrolyte concentrations at the initial moment. The time evolution of CBLs at the membranes surfaces causes that the voltages behave differently depending on the type of membrane system and thus the way of NaCl diffusion in solutions and hydrodynamic instabilities in the chambers caused by gravity. The effect of this is the difference between the final voltage values in these systems in steady states.

The next figure shows the voltages determined in one- and two-membrane systems in steady states as a function of higher NaCl concentrations at the initial moment. Voltages in steady states were taken as the observed voltages after 300 minutes of CBLs reconstruction. In

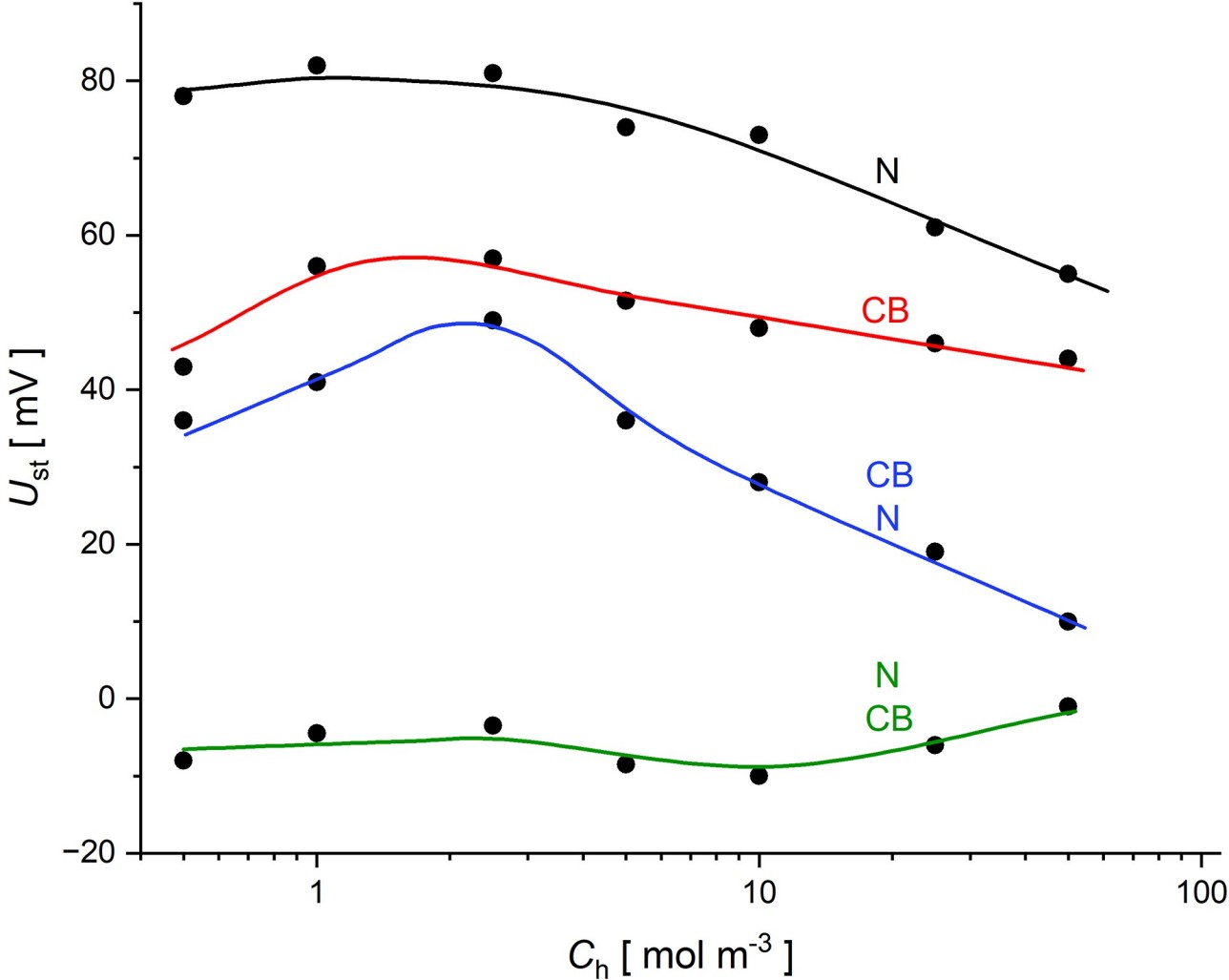

**Fig 3. Steady-state voltages for one- and two-membrane systems with aqueous NaCl solutions as a function of higher NaCl concentration at the initial moment ($C_h$), for a one-membrane system with:** *Nephrophan* **membrane (N–black),** *Biofill* **membrane (CB—red) and for a two-membrane system with configurations:** *Biofill* **as lower membrane (green);** *Nephrophan* **as lower membrane (blue).**

most cases, the voltage pulsations are small after this time of CBLs reconstructing, and therefore the average voltage values in the same short time intervals (10 minutes) at 300 minutes were taken as the established value. Fig 3 shows the steady-state voltages for one-membrane systems in configurations with a solution of higher concentration above the membrane *Nephrophan* (black), *Biofill* (red) and two-membrane systems in configurations with a solution of higher concentration between the membranes and with upper *Nephrophan* membrane (green) and upper *Biofill* membrane (blue).

As is shown in Fig 3, the voltages established as a result of mutually opposed processes of CBLs diffusive reconstruction and convective blurring of CBLs are greater for the *Nephrophan* membrane. This results from higher initial voltages for this membrane system compared to the system with *Biofill* membrane and lower transport properties of *Nephrophan* membrane compared to *Biofill* membrane. Voltages after 300 minutes are different from zero in the entire range of the tested initial NaCl concentrations, which proves the heterogeneity of the solutions

in the membrane system after this time and potential possibility of further processes that equalize the concentrations in the system and reduce the existing thermodynamic forces supporting these processes. On the other hand, in the case of the two-membrane system with the *Nephrophan* and *Biofill* membranes, the diffusion of the solution from the middle chamber through both membranes results in a faster reduction of thermodynamic forces on the membranes compared to the one-membrane system. The result of this, and lower voltages in the two-membrane system at the initial moment, are lower voltages in the steady states of two-membrane system compared to one-membrane systems.

In the case of two-membrane systems, asymmetry can be observed due to the arrangement of the membranes in relation to the vector of gravitational field. When the *Biofill* membrane is in the lower location, the voltage in the steady states settles to values close to zero over the entire range of the initial NaCl concentrations. On the other hand, when the *Nephrophan* membrane is in the lower location, the steady-state voltages are much higher than in the case when the *Biofill* membrane is in the lower location, especially for lower values of the initial NaCl concentration. In the two-membrane system with binary solutions, hydrodynamic instabilities appear only at the lower membrane, so the greater intensity of convection processes occurring at the *Biofill* membrane leads to a faster equalization of concentrations at the electrode surfaces, and thus to lower steady-state voltages observed in the system.

In the two-membrane system, it can be expected that the system of symmetrically arranged electrodes with respect to the membranes and immersed in solutions with lower electrolyte concentrations will be more sensitive to changes in concentrations of the electrolyte. The observed time required for increasing in CBL thickness to appear hydrodynamic instabilities in a two-membrane system is shorter (higher $T_p^{-1}$ values) than in one-membrane systems [13, 21]. On the other hand, for ternary solutions (aqueous NaCl and ethanol solutions) in a two-membrane system, hydrodynamic instabilities may appear at the lower or upper membrane. If, at the initial moment, the density of the solution in the chamber between the membranes is higher than the densities of the solutions in the outer chambers, the CBLs reconstruction may lead to the appearance of hydrodynamic instabilities at the lower membrane. However, if the density of the solution in the chamber between the membranes is initially lower than in the outer chambers, the processes occurring near the membranes may lead to the appearance of hydrodynamic instabilities near the upper membrane.

Fig 4 shows the time characteristics of the voltages measured in a two-membrane system with ternary solutions (aqueous NaCl and ethanol solutions) for the same initial NaCl concentration in the middle chamber, equal to $C_h = 10$ mol m$^{-3}$ (in the outer chambers, $C_l = 0.01$ mol m$^{-3}$) and different initial ethanol concentrations (in the outer chambers, the ethanol concentration at the initial moment was zero). In subsequent graphs from 4a to 4h, the concentration of ethanol at the initial moment was gradually increased from 0 to 90 mol m$^{-3}$. For NaCl concentration at initial moment in the middle chamber $C_h = 10$ mol m$^{-3}$, the concentration of ethanol, at which the density of the solution in the middle chamber was approximately equal to the densities of solutions in the outer chambers, is about 50 mol m$^{-3}$. The measured initial voltages in all cases were in the range of 58–61 mV.

As can be seen in Fig 4 the initial ethanol concentrations do not influence significantly the initial voltages in the two-membrane system, and the way of evolution of CBLs in a diffusive way. In this range of time, the graphs for both configurations of two-membrane systems coincide. The emergence of hydrodynamic instabilities in two-membrane systems causes a "bifurcation" of voltage characteristics due to the configuration of the system, i.e. depending on which membrane is the lower membrane. For the initial density in the middle chamber greater than the densities of the solutions in the outer chambers, (graphs 4a to 4d), hydrodynamic instabilities appear at the lower membrane. On the other hand, in the case of initial density of

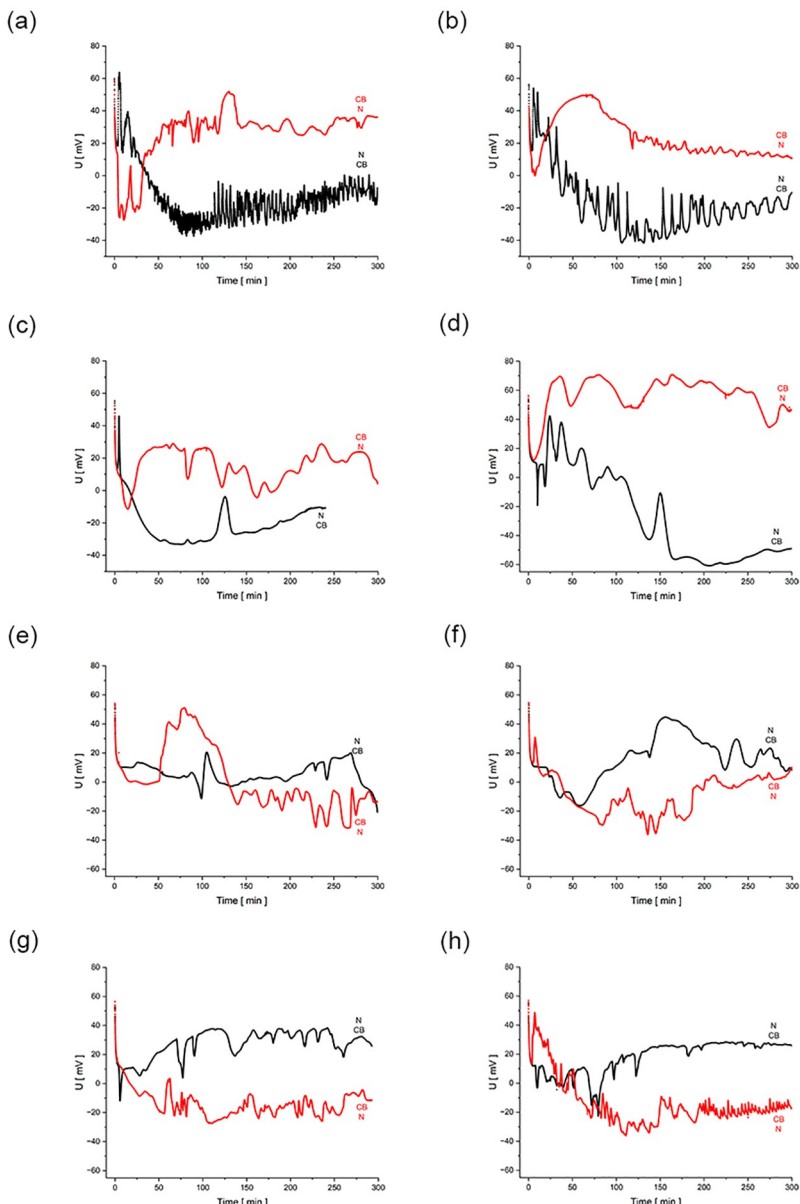

**Fig 4. Time characteristics of voltages in two-membrane systems with aqueous NaCl solutions, for different initial concentrations of ethanol in the middle chamber and with lower membrane *Biofill* (black) and with lower membrane *Nephrophan* (red).** Initial concentration of NaCl in the middle chamber $C_h$ = 10 mol m$^{-3}$. The ethanol concentrations in the middle chamber at the initial moment was equal to: a) 0 mol m$^{-3}$, b) 20 mol m$^{-3}$, c) 40 mol m$^{-3}$, d) 50 mol m$^{-3}$, e) 55 mol m$^{-3}$, f) 60 mol m$^{-3}$, g) 70 mol m$^{-3}$ and h) 90 mol m$^{-3}$.

solution in the middle chamber lower than solutions densities in the outer chambers (graphs 4e to 4h), hydrodynamic instabilities appear at the upper membrane. Considering the membrane at which hydrodynamic instabilities are observed, when hydrodynamic instabilities appear at the *Biofill* membrane (black graphs—Fig 4a–4d, red graphs—Fig 4e–4h), changes in the average voltage are observed towards lower values than when instabilities are observed near the *Nephrophan* membrane (red graphs—Fig 4a–4d, black graphs—Fig 4e–4h). This is due to the greater intensity of hydrodynamic instabilities at the *Biofill* membrane (a greater

amplitude and frequency of observed voltage pulsations). The *Biofill* membrane has better transport properties than the *Nephrophan* membrane, and thus near the *Biofill* membrane, CBLs are rebuilt by diffusion faster compared to the case of *Nephrophan* membrane.

The character of voltage pulsations of the measured voltage in the two-membrane system, associated with hydrodynamic instabilities in the vicinity of the membranes, depends on the dynamics of hydrodynamic instabilities. The intensity of hydrodynamic instabilities can be related to the amplitude and average frequency of pulsations [13]. The critical Rayleigh number above which the pulsations are observed separates the diffusion and diffusive-convective ranges of CBLs regeneration. As can be seen from the graphs in Fig 4, a greater "distance" from the ethanol concentration in the middle chamber of the system, at which the densities in all chambers of the two-membrane system are the same, causes that observed characteristics are characterized by higher frequencies and amplitudes of pulsations, which indicates a greater intensity of free convection. This can be related to higher Rayleigh numbers for CBLs, which may also indicate a different type of stirring of the solution by free convection, leading to different spatial structures, e.g. "finger type" or "plum structure". As it results from earlier analyses of pulsations [12], they are also chaotic in nature.

Fig 5 shows the voltages in two-membrane system with membranes in horizontal planes in a steady state after six hours of CBLs reconstruction near the membrane surfaces, for ternary solutions with the same initial NaCl concentration ($C_h = 10$ mol m$^{-3}$) and different ethanol concentrations in the middle chamber.

The steady-state voltages observed in two-membrane systems with ternary solutions show a strong dependence on the concentration of ethanol, used in this case as a solution density modifier, and thus influencing density gradients in CBLs. The steady-state voltages for both configurations of two-membrane systems vary over entire range of ethanol concentrations used. These relationships are characterized by a transition between two ranges of concentrations: lower than 40 mol m$^{-3}$ and greater than 60 mol m$^{-3}$. When the concentration of ethanol in the middle chamber is equal to 50 mol m$^{-3}$, the initial densities of the solutions in the chambers are the same. This results from Eq (11) with assumption that the NaCl concentration at the initial moment in the middle chamber was equal to 10 mol m$^{-3}$. In the range of transient ethanol concentrations (between 40 and 60 mol m$^{-3}$), the maximum of voltages in steady states is observed for one configuration of the two-membrane system (with the lower *Nephrophan* membrane) and the minimum for the second configuration of the two-membrane system (with the lower *Biofill* membrane). Both observed extremes occur for the concentration of ethanol, for which the initial densities of the solutions in the chambers are the same. Large voltage differences in the extremes indicate lower intensity of hydrodynamic instabilities in the transition range of ethanol concentrations and increased sensitivity of the membrane system to gravity in this range of ethanol concentrations. Analysing the graphs in Fig 5, it can be concluded that if hydrodynamic instabilities appear near the *Biofill* membrane (black graph—for low ethanol concentrations and red graph—for high ethanol concentrations in the middle chamber at the initial moment), the steady-state voltages are lower than steady-state voltages measured when hydrodynamic instabilities occur near the *Nephrophan* membrane.

By analyzing the voltage-time characteristics (on a logarithmic scale), the characteristic time ($T_p$) required for rebuilding enough the density gradients in the CBLs to initiate hydrodynamic instability processes can be determined. In voltage-time graphs, this is visible as the beginning of a periodic changes of voltage in time. This is one of the parameters that allow better characterization of the interaction of diffusion with hydrodynamic instabilities (natural convection) in the membrane systems. This time can be defined as: $T_p$—the time from the moment of switching off the mechanical stirring of solutions (ensuring their homogeneity) to the moment of a visible deviation of the measured potential difference between Ag|AgCl

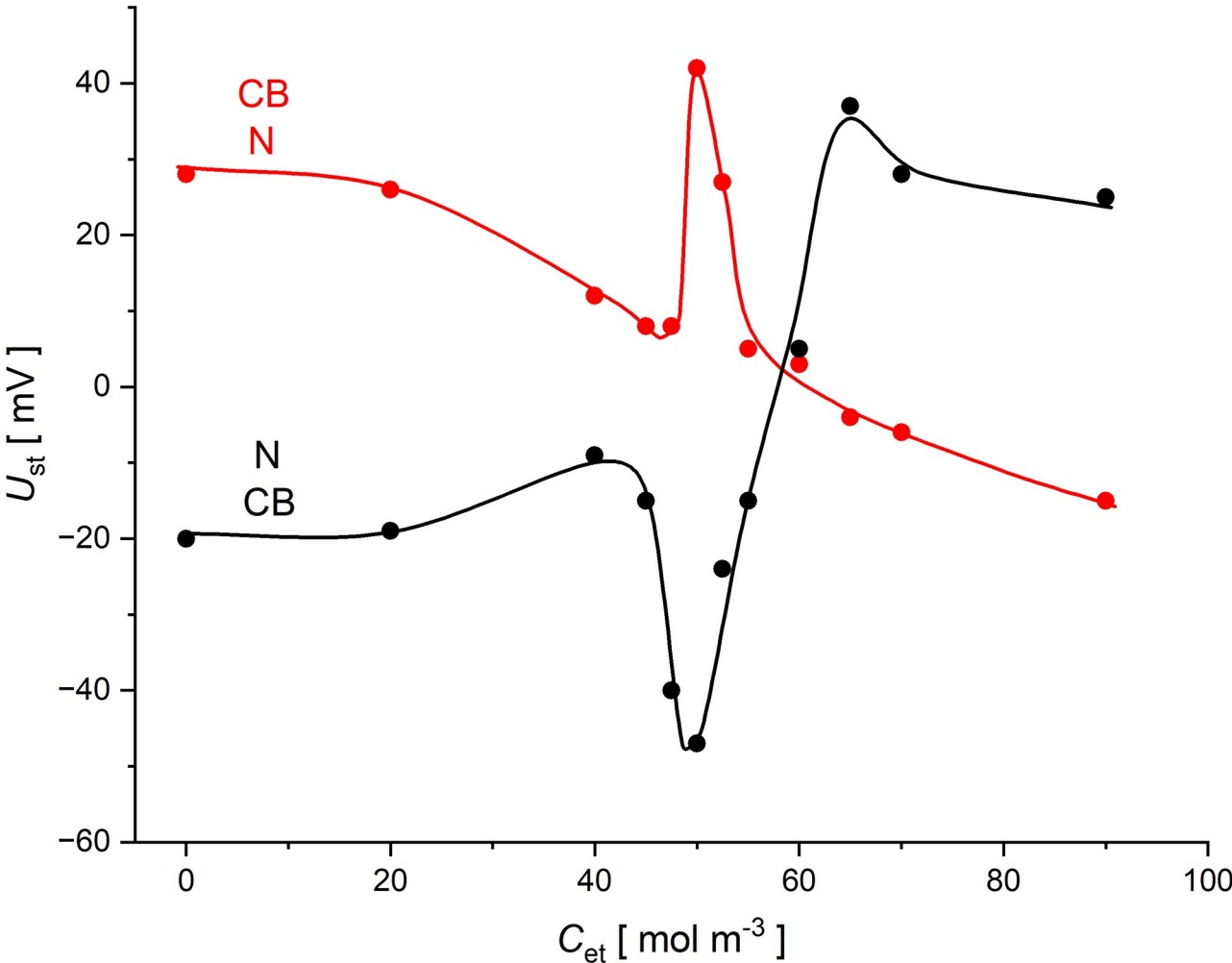

**Fig 5. Steady-state voltages for a two-membrane system as a function of initial ethanol concentration in the middle chamber, for the system with lower *Nephrophan* membrane (red) and lower *Biofill* membrane (black).** The NaCl concentration in the middle chamber at the initial moment was equal to 10 mol m$^{-3}$.

electrodes from the monotonic and nonlinear decrease of the voltage between electrodes, which is the beginning of voltage pulsation related to hydrodynamic instabilities.

Fig 6 shows the inverse of time needed for appearance of hydrodynamic instabilities in the two-membrane systems as a function of the initial NaCl concentrations without ethanol in the middle chamber of two-membrane system.

As can be seen in Fig 6, similarly as in the case of one-membrane systems in the configuration in which hydrodynamic instabilities appear [12], increase in NaCl concentration in the middle chamber of the two-membrane system shortens the time needed for the appearance of hydrodynamic instabilities (increase of $T_p^{-1}$). In the two-membrane system with binary solutions, instabilities can only appear near the lower membrane. In the case of the two-membrane system, for NaCl concentrations lower than 10 mol m$^{-3}$ at the initial moment, there are no significant differences between the configurations with the *Nephrophan* membrane or the *Biofill* membrane as a lower membrane. These differences are significant at higher concentrations of NaCl in the middle chamber at the initial moment. This

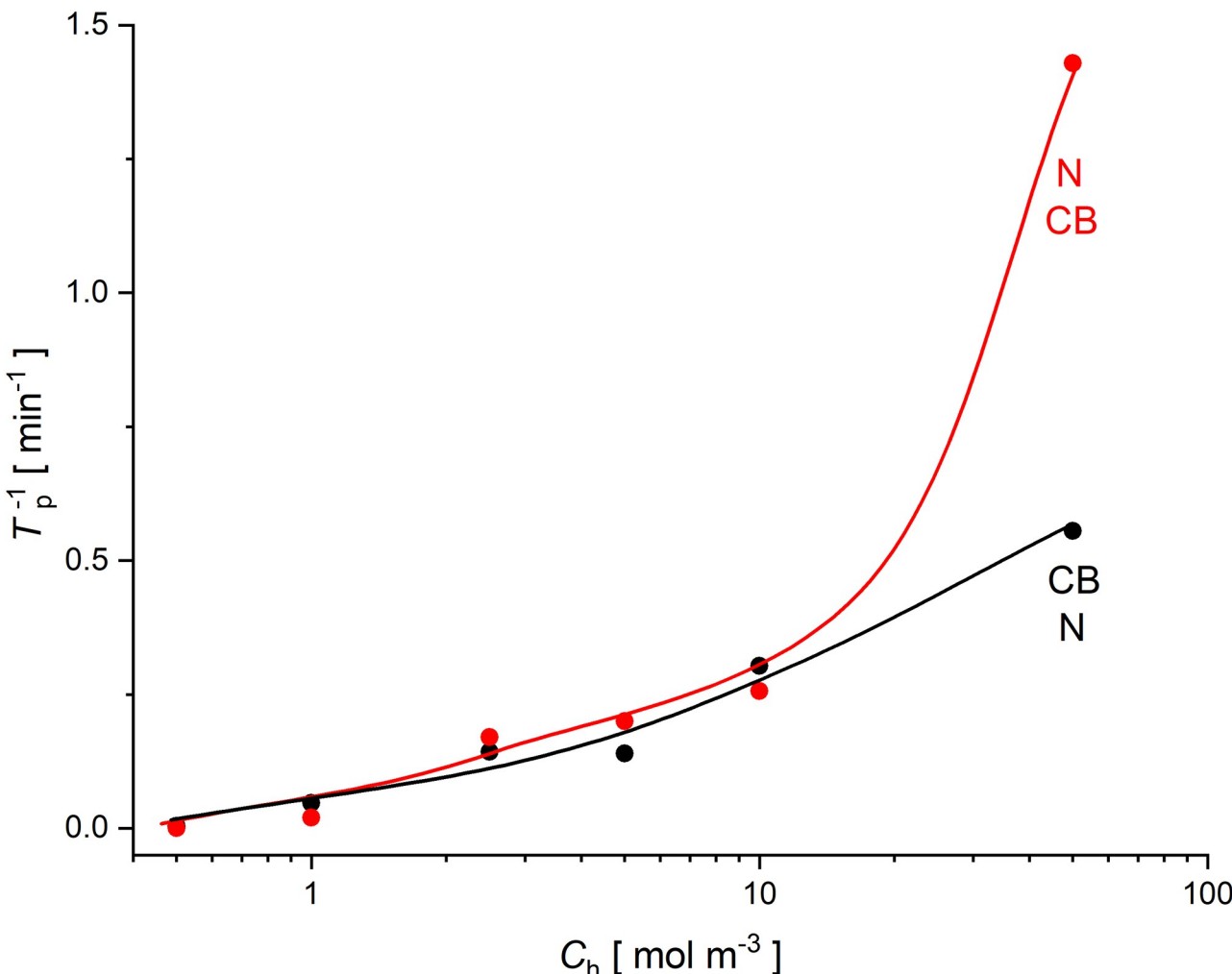

**Fig 6. Inverse of time needed for appearance of hydrodynamic instabilities in the two-membrane system as a function of initial NaCl concentration in the middle chamber without ethanol, for the configuration with lower *Biofill* membrane (red) and lower *Nephrophan* membrane (black).**

may be caused by the fact that the *Biofill* membrane has better permeability of both solvent and solutes through the membrane compared to the *Nephrophan* membrane. When the *Biofill* is the lower membrane, CBLs are reconstructed faster near this membrane and therefore hydrodynamic instabilities may occur earlier than in the case when the *Nephrophan* membrane is the lower membrane.

Fig 7 shows the invers of time needed for appearance of the gravity-induced hydrodynamic instabilities as a function of the initial ethanol concentration in the middle chamber of two-membrane system with *Biofill* (CB) and *Nephrophan* (N) membranes. The initial NaCl concentration in the middle chamber was 10 mol m$^{-3}$. Markers (•) and the line (—) show the results obtained from the calculations based on the model of Eqs (5)–(7), (9) and (10) with critical Rayleigh number $(R_a)_c = 1100.6$, while the markers (○) show the results obtained from the experiment, for the configuration of the two-membrane system: a) with upper *Biofill* membrane and lower *Nephrophan* membrane, while in the case b) the membranes locations were reversed.

(a)

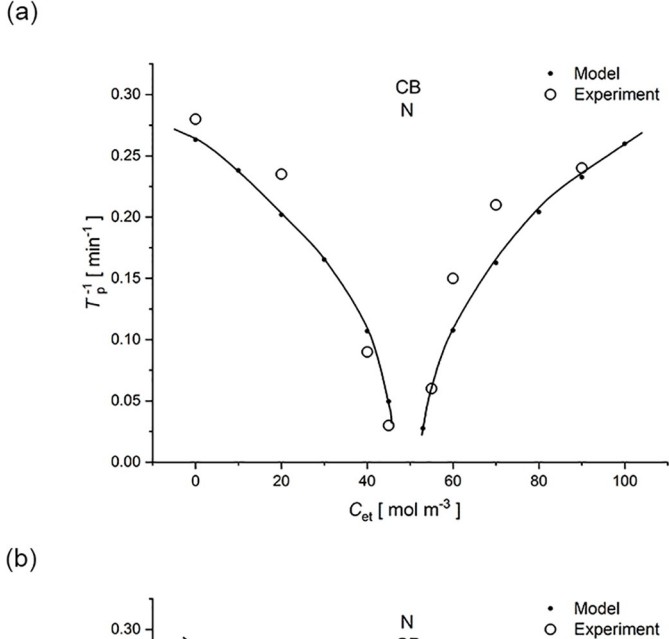

(b)

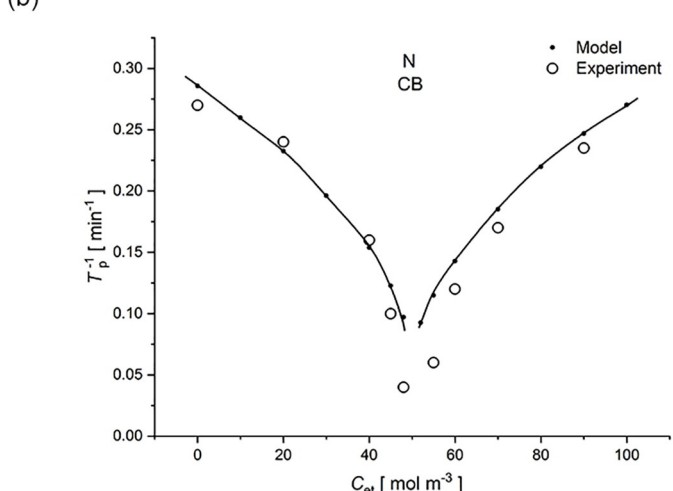

**Fig 7. The inverse of time needed for appearance of hydrodynamic instabilities in a two-membrane system as a function of ethanol concentration in the middle chamber, with the lower *Nephrophan* membrane (a) and the lower *Biofill* membrane (b) with initial NaCl concentration in the middle chamber 10 mol m$^{-3}$, lines and full points (•)—model, markers without filling inside (○)—experiment.**

It should be noted that in the case of low ethanol concentrations in the middle chamber at the initial moment ($C_{et}$ smaller than 50 mol m$^{-3}$, densities of solutions in the middle chamber at the initial moment are higher than in the outer chambers), hydrodynamic instabilities appear at the lower membrane. On the other hand, when the concentration of ethanol in the middle chamber at the initial moment is higher than 50 mol m$^{-3}$, hydrodynamic instabilities appear at the upper membrane. A characteristic feature of both the experimental results and the model is that the highest values of times needed for appearance of hydrodynamic instabilities (the lowest values of $T_p^{-1}$) were obtained in the range of ethanol concentrations, for which the solutions in the middle chamber and outer chambers had similar densities. For NaCl concentration in the middle chamber at the initial moment equal to 10 mol m$^{-3}$, the concentration of ethanol at which the densities of solutions in the chambers are equal at the initial moment,

according to the Eq (11) is 50 mol m$^{-3}$. The more the ethanol concentration at the initial moment in the middle chamber differs from the concentration 50 mol m$^{-3}$, both towards lower and higher ethanol concentrations, the more $T_p^{-1}$ increases in a non-linear way ($T_p$ decreases because hydrodynamic instabilities appear earlier). A certain symmetry of these curves can be observed with respect to the ethanol concentration at the initial moment in the middle chamber equal to 50 mol m$^{-3}$. Comparing the systems with the *Biofill* membrane as a lower membrane with the *Nephrophan* membrane as a lower membrane, it can be concluded from Fig 7 that $T_p^{-1}$ for the system with the lower *Biofill* membrane are slightly higher than those with the lower *Nephrophan* membrane. This is connected with the better transport properties of the *Biofill* membrane compared to the *Nephrophan* membrane, which results in faster formation of CBLs at the *Biofill* membrane. Considering experimental points–open circles in Fig 7b we compare symmetric ethanol concentrations with respect to 50 mol/m$^3$ (for which the initial solution densities in all chambers are the same), i.e. 40 mol/m$^3$ and 60 mol/m$^3$. It can be stated that for the initial ethanol concentration in the middle chamber of 40 mol/m$^3$, the solution density in the middle chamber was initially higher than in the outer chambers, which caused hydrodynamic instabilities to appear at the lower membrane (i.e. the *Biofill* membrane). The time needed for the voltage pulsation to appear in this case was shorter than in the case of the initial ethanol concentration in middle chamber of 60 mol/m$^3$ ($t_p^{-1}(40 \text{ mol/m}^3) > t_p^{-1}(60 \text{ mol/m}^3)$), for which the initial density in the middle chamber was lower than in the outer chambers, which in this case caused the occurrence of instability at the upper membrane (*Nephrophan* membrane).

An analogous comparison for the case presented in Fig 7a gives: $t_p^{-1}(40 \text{ mol/m}^3) < t_p^{-1}(60 \text{ mol/m}^3)$ i.e. the time of occurrence of instability at the lower membrane (*Nephrophan*–the case of initial ethanol concentration of 40 mol/m$^3$) is longer than when instabilities appear at the upper membrane (*Biofill* membrane–the case of initial ethanol concentration of 60 mol/m$^3$), which could be expected. This difference is small on the theoretical curves.

Fig 8 shows the inverse of time needed for appearance of gravity-induced hydrodynamic instabilities ($T_p$) as a function of initial ethanol concentration for a two-membrane system with *Biofill* (CB) and *Nephrophan* (N) membranes. The initial NaCl concentrations in the middle chamber were: 5 mol m$^{-3}$ (blue), 10 mol m$^{-3}$ (red) and 15 mol m$^{-3}$ (black). The times $T_p$ were calculated from the model, for the system configuration with the *Nephrophan* as a lower membrane (a) and the *Biofill* as a lower membrane (b).

For different NaCl concentrations at the initial moment in the middle chamber, the characteristics $T_p^{-1} = f(C_{et})|_{C_{\text{NaCl}=\text{const.}}}$ are similar, but shifted relative to each other so that the minimum of the curves, for a given initial NaCl concentration, is observed for the ethanol concentration, at which the densities of the solutions in the chambers of the two-membrane system at the initial moment, according to Eq (11), are the same. This is consistent with both the two-membrane systems with the *Biofill* as an upper membrane (a) as well as a lower membrane (b).

## Conclusions

1. The model of CBLs recovery at the membrane has been extended to two-membrane systems with ternary solutions

2. The voltages in one-membrane systems at the initial moment (homogeneous, binary solutions in the chambers) are higher than in the case of two-membrane systems and depend on the type of the membrane. The initial voltage for a two-membrane system can be

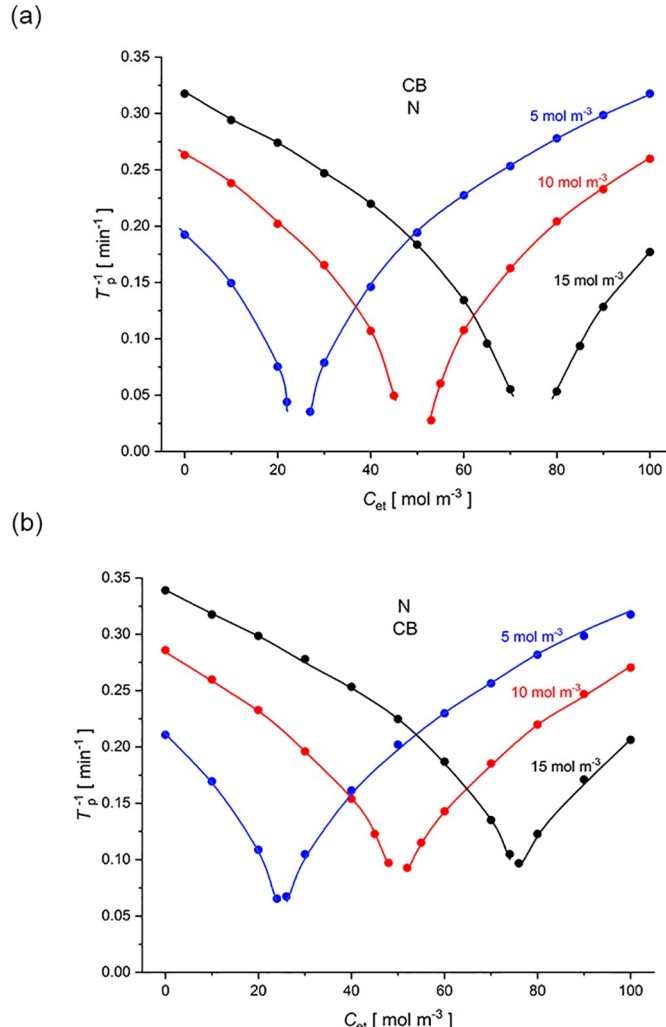

**Fig 8. The inverse of time needed for the appearance of hydrodynamic instabilities in the two-membrane system as a function of the ethanol concentration in the middle chamber at the initial moment, calculated from the model.** The initial NaCl concentrations in the middle chamber were: 5 mol m$^{-3}$, 10 mol m$^{-3}$ and 15 mol m$^{-3}$. Configuration of the two-membrane system: with the *Nephrophan* as a lower membrane (a) and the *Biofill* as a lower membrane (b).

calculated as the difference of the respective voltages obtained in the individual one-membrane systems

3. The steady-state voltages for one-membrane systems are also greater than steady-state voltages for two-membrane systems with suitable membranes, however, the steady-state voltages of the two-membrane systems cannot be calculated from the steady-state voltages of single-membrane systems. Comparing the voltages established in the two-membrane system with NaCl aqueous solutions, it can be concluded that the voltages established for the system with the *Biofill* as a lower membrane are significantly lower compared to the system with the *Nephrophan* as a lower membrane. The reason is the better transport properties of the *Biofill* membrane

4. The voltage characteristics of two-membrane systems differ due to: the localization of membranes in the system and the initial NaCl and ethanol concentrations in the middle chamber. In the case of a fixed initial NaCl concentration in the middle chamber, for low ethanol concentrations, the cause of hydrodynamic instabilities is the difference of solutions densities at the lower membrane, while for high ethanol concentrations, the cause of hydrodynamic instabilities is the difference of solutions densities at the upper membrane. The ethanol concentration between these concentration ranges is the ethanol concentration for which, at the initial moment, the densities of the solutions in the chambers of the two-membrane system are the same

5. The steady-state voltage of the two-membrane system as a function of the initial ethanol concentration with the same initial NaCl concentration in the middle chamber is a complex function depending on the localization of membranes in two-membrane system, with characteristic transition near the ethanol concentration, for which the initial densities of solutions in the chambers of the two-membrane system are the same

6. In the two-membrane system and aqueous NaCl solutions, the inverse of time needed for appearance of the hydrodynamic instabilities as a function of NaCl concentration is a non-linear, increasing function, weakly dependent on the NaCl concentration for both configurations of the two-membrane system in the low concentration range, while for NaCl concentrations higher than 10 mol m$^{-3}$ the differences are significant

7. The time required to the appearance of hydrodynamic instabilities in a two-membrane system and aqueous NaCl and ethanol solutions depends on the initial ethanol concentration in the middle chamber, with the assumption of the same initial NaCl concentration in the middle chamber. This relationship is similar for both configurations of the two-membrane system and consists of two non-linear curves converging to the ethanol concentration, at which the densities of the solutions at the initial moment in all chambers of the two-membrane system are the same.

## Supporting information

**S1 File. Raw data for figures.**
(XLSX)

## Author Contributions

**Conceptualization:** Sławomir Grzegorczyn, Andrzej Ślęzak.

**Data curation:** Sławomir Grzegorczyn, Iwona Dylong, Paweł Dolibog.

**Formal analysis:** Sławomir Grzegorczyn, Andrzej Ślęzak.

**Funding acquisition:** Sławomir Grzegorczyn, Iwona Dylong.

**Investigation:** Sławomir Grzegorczyn, Iwona Dylong.

**Methodology:** Sławomir Grzegorczyn, Andrzej Ślęzak.

**Project administration:** Sławomir Grzegorczyn.

**Resources:** Sławomir Grzegorczyn.

**Software:** Sławomir Grzegorczyn.

**Supervision:** Sławomir Grzegorczyn, Andrzej Ślęzak.

**Validation:** Sławomir Grzegorczyn, Andrzej Ślęzak.

**Visualization:** Sławomir Grzegorczyn, Paweł Dolibog.

**Writing – original draft:** Sławomir Grzegorczyn, Iwona Dylong.

**Writing – review & editing:** Sławomir Grzegorczyn, Paweł Dolibog, Andrzej Ślęzak.

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
