## [Decision Letter · Decision Letter 0]

14 Oct 2024

PONE-D-24-37753Diffusion and hydrodynamic instabilities in membrane systems with water solutions of NaCl and ethanolPLOS ONE

Dear Dr. Grzegorczyn,

Thank you for submitting your manuscript to PLOS ONE. After careful consideration, we feel that it has merit but does not fully meet PLOS ONE’s publication criteria as it currently stands. Therefore, we invite you to submit a revised version of the manuscript that addresses the points raised during the review process.

We look forward to receiving your revised manuscript.

Kind regards,

Mallikarjuna Reddy Kesama, Ph.D.

Academic Editor

PLOS ONE

Journal Requirements:

Additional Editor Comments:

Major Revision

Reviewers' comments:

Reviewer's Responses to Questions

**Comments to the Author**

1. Is the manuscript technically sound, and do the data support the conclusions?

Reviewer #1: Yes

2. Has the statistical analysis been performed appropriately and rigorously? 

Reviewer #1: N/A

3. Have the authors made all data underlying the findings in their manuscript fully available?

Reviewer #1: Yes

4. Is the manuscript presented in an intelligible fashion and written in standard English?

Reviewer #1: Yes

5. Review Comments to the Author

Reviewer #1: The work by Grzegorczyn et al. studies the impact of hydrodynamic instability in the transport of chemical species across a membrane. The resulting effects are investigated by measuring the difference of voltage between the separated solutions in one- and two-membrane systems. The influence of density gradients and membrane properties are taken into account as critical parameters to control the system dynamics. Experimental data favorably compare with trends obtained by using the Kedem-Katchalsky model coupled to diffusion equation. The study aims at extending previous work made on single-membrane systems to two-membrane configurations.

In general, the work is carried out in a consistent and professional way. I found the writing a bit wordiness. Publication is recommended after the following points are properly addressed.

- Introduction

Pag. 3, line 49-51: This discussion should mention that density gradients can also occur and be controlled in-situ due to chemical reactions (see Physics of Fluids 25 (1), 2013, The Journal of Physical Chemistry Letters 5 (5), 875-881, 2014);

Pag. 5, line 93-114: I found hard time to find the link between this study and cancer therapies. I understand the possible impact of gravitation in that kind of studies. However either the author substantiate the actual role of the gravitational instabilities considered at the cellular level or I would shorten this discussion to more concrete information.

- Theory

-Line 135: Missing word? “suitably …”

-z_s in equation 2 not defined,

-\\zeta is not defined

-The authors should explain more explicitly how they compute the inverse time for the onset of the hydrodynamic instability, which is then used for the characterization in Figs. 6, 7, 8.

- Materials and methods

Fig. 1: I don’t understand what the double lines in the middle of each reservoirs are describing.

- Results and Discussion

- Beginning of Results and Discussion, repeats what already said in Materials and methods.

-Could the authors comment on the non-monotonic profile shown by the system “N” in Fig. 2? (i.e. I would expect a monotonic increase with the salt concentration)

-The authors qualitatively explain the onset of voltage pulsations as a manifestation of natural convection. However one could also expect turbulent or erratic fluctuations due to this phenomenon. Indeed the presence of periodic regimes shown for certain density conditions is not trivial and very interesting. Can the authors characterize and discuss more in detail these dynamics (parametric range, properties and originating mechanism) ?

Figs. 7-8: I would expect an asymmetric behavior of the branches with respect to the iso-density: being the transport through the CB membrane faster, the inverse time of the corresponding hydrodynamic instability should be higher than that of N (if the density gradient is comparable). In some cases this is observed but in other the reverse trend in obtained. Could the authors comment on this?

6. PLOS authors have the option to publish the peer review history of their article (what does this mean?). If published, this will include your full peer review and any attached files.

Reviewer #1: No

---

## [Author Response · Author response to Decision Letter 0]

19 Nov 2024

I also included a response to reviewers in the file "Response to Reviewers.docx" ( with appropriate colors for the reviewer's and author's text):

PONE-D-24-37753

Diffusion and hydrodynamic instabilities in membrane systems with water solutions of NaCl and ethanol

PLOS ONE

Reviewer #1:

The work by Grzegorczyn et al. studies the impact of hydrodynamic instability in the transport of chemical species across a membrane. The resulting effects are investigated by measuring the difference of voltage between the separated solutions in one- and two-membrane systems. The influence of density gradients and membrane properties are taken into account as critical parameters to control the system dynamics. Experimental data favorably compare with trends obtained by using the Kedem-Katchalsky model coupled to diffusion equation. The study aims at extending previous work made on single-membrane systems to two-membrane configurations. 

In general, the work is carried out in a consistent and professional way. 

Author's Response to Reviewer

Thank you for your important and interesting review of our article. The reviewer's comments not only allowed us to improve the article but also allowed us to look at some problems in a way that we had not considered before. We tried to take into account the reviewer's advice and looked critically at the presented topic in order to improve the way of information presentation in a concise way.

Rev1: I found the writing a bit wordiness. Publication is recommended after the following points are properly addressed.

We analyzed our article and tried to remove unnecessary fragments of text from the point of view of the essence of the presented problem. 

- Introduction

Rev1: Pag. 3, line 49-51: This discussion should mention that density gradients can also occur and be controlled in-situ due to chemical reactions (see Physics of Fluids 25 (1), 2013, The Journal of Physical Chemistry Letters 5 (5), 875-881, 2014);

This is an important comment from the reviewer. In the previous studies [J. Non-Equilibrium Thermodyn. 2012;37(1):77–99, J. Porous Media 2020;23(4):425–44, Plos One 2022;17(2): e0263059], we paid attention to free convection caused by concentration gradients of electrolytic substances, which caused the formation of density gradients in the vicinity of the membranes. This allowed us to select a method of measuring the potentials between Ag|AgCl electrodes for imaging the processes of diffusional rebuilding of CBLs and their blurring by hydrodynamic instability processes – free convection in CBLs and outside them. In fact, we focused on this type of aspects of the Rayleigh–Taylor instability, assuming no temperature difference in the measurement system. Our considerations for extending of interpretation of the observed hydrodynamic instabilities, especially in relation to biological systems, concerned the possibility of this type of instability to occur in a biological system. Such hydrodynamic instabilities could affect transport conditions in cells or tissues. For this reason, microgravity, in which free convection does not occur, would be one of the potential disturbances of biological systems, through the possibility of disturbing the transport of substances in the organism. We are also aware that the Rayleigh number, strongly depends on the size of the system (l^3), in which free convection as a symptom of hydrodynamic instability may appear. Therefore, since the cell size is in the order of micrometers, hydrodynamic instability caused by density differences in the cell area is unlikely to occur. However, the hydrodynamic instabilities indicated by the Reviewer, caused by chemical reactions at the interface of contacting reacting substances significantly expand the possibilities of interpreting of hydrodynamic instabilities in biological systems and their influence on transport processes. For this reason, in accordance with the reviewer's suggestions, we limited the part of the introduction regarding the potential impact of hydrodynamic instabilities on cancer (to specified citations of works) and we include in the article short description of the appearance of hydrodynamic instabilities related to reactions that disturb densities in solutions. An important effect that we did not emphasize in our studies is the symmetry of instability formation in membrane systems. We also included this issue in the appropriate part of the article in accordance with the reviewer's suggestion.

Rev1: Pag. 5, line 93-114: I found hard time to find the link between this study and cancer therapies. I understand the possible impact of gravitation in that kind of studies. However either the author substantiate the actual role of the gravitational instabilities considered at the cellular level or I would shorten this discussion to more concrete information. 

We limited the part of the introduction regarding the potential impact of instability on cancer (to few important citations of works) and include in the article the appearance of hydrodynamic instabilities related to the chemical reactions disturbing the density in solutions. It seems that in the previous version of the article we did not dedicate enough attention to this important issue (reactive diffusive instability), especially in relation to the biological system, focusing only on hydrodynamic instabilities without reactions.

- Theory

Rev1: -Line 135: Missing word? “suitably …”

The comment was taken into account and the sentence was corrected.

Rev1: -z_s in equation 2 not defined,

zs - was defined in the article

Rev1: -\\zeta is not defined

 \\zeta - is defined in the article 

Rev1: -The authors should explain more explicitly how they compute the inverse time for the onset of the hydrodynamic instability, which is then used for the characterization in Figs. 6, 7, 8.

The definition of the time at which hydrodynamic instabilities appear in a membrane system with the membrane in the horizontal plane and with a solution of higher density above the membrane have appeared in previous articles [J. Non-Equilibrium Thermodyn. 2012;37(1):77–99]. In this case we did not avoid a mental shortcut, and after analysis we tried to determine more precisely this important parameter related to hydrodynamic instabilities. We described this time and its definition in the section “Results and discussion” (before the Fig. 6, at which it appeared first time). 

T_p - this is the time from the moment of switching off the mechanical stirring of solutions (ensuring their homogeneity) to the moment of a visible deviation of the measured potential difference between Ag|AgCl electrodes from the monotonic and non-linear decrease of the voltage between electrodes, which is the beginning of voltage pulsation related to hydrodynamic instabilities. 

- Materials and methods 

Rev1: Fig. 1: I don’t understand what the double lines in the middle of each reservoirs are describing.

The double line in the measuring systems represents a crossbar mounted in the chamber (not restricting the free flow or diffusion of the solution) supporting the stirring rod (cylinder) with stirrer so that the stirring of solutions is most effective near the membrane surface. 

- Results and Discussion 

Rev1: - Beginning of Results and Discussion, repeats what already said in Materials and methods.

We have corrected these parts of the article so that there are no repetitions of the sentences describing measurement procedures. 

Part of the text (after appropriate adaptation) was moved from the "Results and discussion" chapter to "Materials and methods", because it actually fits better into that chapter. 

Rev1: - Could the authors comment on the non-monotonic profile shown by the system “N” in Fig. 2? 

 (i.e. I would expect a monotonic increase with the salt concentration)

The nonlinear dependence of Uo on Ch in Fig. 2 for both the Nephrophan (N) and Bacterial Cellulose (CB) membrane systems and for the system with these two membranes is obtained at the moment of initial observation of the voltage between the electrodes in the membrane systems. This is the moment of still homogeneous solutions in the chambers, without concentration boundary layers (CBLs) at the membranes. In this case, the voltage difference between the electrodes is influenced not only by the concentrations at the electrodes themselves but also by the apparent ion transfer number of the membrane. The relationship between the measured voltage between the electrodes in a membrane system with homogeneous solutions in the chambers is presented by the equation

∆ψ= - 2RT/F ∙ <<t+m>> ∙ ln(ah/al)

where: <<t+m>> - apparent transfer number of positive ions in the membranę, ah, al – ion activity in the chambers of the membrane system. As our previous studies have shown, this number depends on the concentrations of solutions in the chambers of the membrane system (or more precisely, the quotient of concentrations on both sides of the membrane).

From the presented dependencies of Uo on Ch in Fig. 2 it follows (for the points and the curve for the two-membrane system) that by measuring the voltage in the two-membrane system in the initial conditions (homogeneous solutions in the chambers) – black points – they can be composed (predicted) from the voltages on the individual membranes. The voltages are subtracted (<<DeltaPsi>>N - <<DeltaPsi>>BC), because the polarizations of the concentrations on the membranes in the two-membrane system, and consequently the polarizations of the electric fields on the membranes, are directed oppositely. This type of simple composition of the voltages from systems with individual membranes for a multi-membrane system is not fulfilled for steady states, it means for membranes with concentration polarization, (e.g. results presented in Fig. 5 for the two-membrane and systems with ternary solutions). In this case the orientation of the membranes in the two-membrane system with respect to the gravitational field and the diffusion and free convection (caused by hydrodynamic instabilities) of solutions at different membranes, depending on the density distribution of the solutions with respect to the membranes, contributes to the complex characteristics of Uo = f(Ch). 

Rev1: -The authors qualitatively explain the onset of voltage pulsations as a manifestation of natural convection. However one could also expect turbulent or erratic fluctuations due to this phenomenon. Indeed the presence of periodic regimes shown for certain density conditions is not trivial and very interesting. Can the authors characterize and discuss more in detail these dynamics (parametric range, properties and originating mechanism)?

The analysis of voltage pulsations indicates on their chaotic character (Fourier analysis of pulsations [Plos One 2022;17(2): e0263059]), which means that the voltages in analogous steady states for systems with single membrane do not combine into a voltage for a double-membrane system as in the case of homogeneous solutions. A characteristic feature of the observed voltage pulsations (irregular oscillations) in the case of membrane systems (one- and two-membrane systems with binary and ternary solutions) is the decrease in the pulsation amplitude with time, and the different pulsation "frequency" shown by the average number of pulsations per unit of time (pulsation frequency). The average number of pulsations per unit time can be connected to the intensity of free convection associated with hydrodynamic instabilities. As can be seen from the graphs in Figure 4, the lowest "pulsation frequencies" are observed for solutions with similar initial densities in the measuring chambers of the double-membrane system (Fig. 4d). The change in the initial ethanol concentrations in the middle chamber (both towards lower and higher concentrations compared to 50 mol/m3) causes the difference in the initial densities of the solutions in the outer chambers and the middle chamber to increase, which in Figures 4 (in the directions: d - c - b - a, and d - e - f - g – the direction of increasing the initial difference in density between the chambers) is visible by the increasing frequency of voltage pulsation. This indicates an increasing intensity of hydrodynamic instability near the membrane for which the solution above the membrane has a higher density than the solution below the membrane (the case �c�b�a – applies to hydrodynamic instabilities for the lower membrane and the case e - f - g – applies to hydrodynamic instabilities for the upper membrane).

Rev1: Figs. 7-8: I would expect an asymmetric behavior of the branches with respect to the iso-density: being the transport through the CB membrane faster, the inverse time of the corresponding hydrodynamic instability should be higher than that of N (if the density gradient is comparable). In some cases this is observed but in other the reverse trend in obtained. Could the authors comment on this?

Considering Figure 7b (experimental points – open circles) we compare symmetric ethanol concentrations with respect to 50 mol/m3 (for which the initial solution densities in all chambers are the same), i.e. 40 mol/m3 and 60 mol/m3. It can be stated that for the initial ethanol concentration in the middle chamber of 40 mol/m3, the solution density in the middle chamber was initially higher than in the outer chambers, which caused hydrodynamic instabilities to appear at the lower membrane (i.e. the bacterial cellulose membrane). The time needed for the voltage pulsation to appear in this case was shorter than in the case of the initial ethanol concentration in middle chamber of 60 mol/m3 ( t_p^(-1) (40 mol/m3) > t_p^(-1) (60 mol/m3) ), for which the initial density in the middle chamber was lower than in the outer chambers, which in this case caused the occurrence of instability at the upper membrane (Nephrophan membrane). This may be related to the higher permeability coefficient of the bacterial cellulose membrane to NaCl (compared to the Nephrophan membrane NaCl permeability), for which CBLs are rebuilt faster than in the case of the Nephrophan membrane, and therefore large enough density gradients appear in shorter time in the CBLs to "start" the processes of hydrodynamic instability. This effect is much smaller for the obtained theoretical curve, which may be connected to the approximations accepted in the theoretical model. An analogous comparison for the case presented in Figure 7a gives: t_p^(-1) (40 mol/m3) < t_p^(-1) (60 mol/m3) i.e. the time of occurrence of instability at the lower membrane (Nephrophan – the case of initial ethanol concentration of 40 mol/m3) is longer than when instabilities appear at the upper membrane (bacterial cellulose membrane – the case of initial ethanol concentration of 60 mol/m3), which could be expected. This difference is small on the theoretical curves. For larger deviations of initial ethanol concentrations from 50 mol/m3 the differences between the corresponding initial ethanol concentrations (e.g. 20 and 80 mol/m3) are small, which may result from smaller values of observed times t_p .

Thank you once again for your important comments on the article.

---

## [Editor Report · Decision Letter 1]

27 Nov 2024

Diffusion and hydrodynamic instabilities in membrane systems with water solutions of NaCl and ethanol

PONE-D-24-37753R1

Dear Dr. Sławomir Grzegorczyn,

We’re pleased to inform you that your manuscript has been judged scientifically suitable for publication and will be formally accepted for publication once it meets all outstanding technical requirements.

Kind regards,

Mallikarjuna Reddy Kesama, Ph.D.

Academic Editor

PLOS ONE

---

## [Editor Report · Acceptance letter]

3 Dec 2024

PONE-D-24-37753R1 

PLOS ONE

Dear Dr. Grzegorczyn, 

I'm pleased to inform you that your manuscript has been deemed suitable for publication in PLOS ONE. Congratulations! Your manuscript is now being handed over to our production team.

Kind regards, 

on behalf of

Dr. Mallikarjuna Reddy Kesama 

Academic Editor

PLOS ONE